# Environmental Impact Modeling for Transportation of Hazardous Liquids

**Zdenek Dvorak [1],\***, **Bohus Leitner [1]** 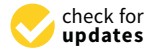, **Michal Ballay [1]**, **Lenka Mocova [2]** and **Pavel Fuchs [3]**

1 Department of Technical Sciences and Informatics, Faculty of Security Engineering, University of Zilina, 01026 Zilina, Slovakia; bohus.leitner@uniza.sk (B.L.); michal.ballay@uniza.sk (M.B.)
2 Institute of Lifelong Learning, University of Zilina, 01026 Zilina, Slovakia; lenka.mocova@uniza.sk
3 Department of Security Technologies and Engineering, Faculty of Transportation Sciences, Czech Technical University in Prague, 16000 Prague, Czech Republic; pavel.fuchs1@gmail.com
* Correspondence: zdenek.dvorak@uniza.sk; Tel.: +421-41-513-6854

**Abstract:** Modeling the effects of leakage in the transport of hazardous liquids is a highly topical issue, not only in the field of environmental engineering. This article's introduction presents relevant information and statistical sources, analyzes selected scientific and professional publications, and characterizes the results of selected research projects. The applied approaches, methods, and results of our research specify the processes of developing and testing a theoretical model of spreading the impacts of leakage of hazardous liquids on biological components of the environment. The proposed model for predicting the environmental impacts of hazardous liquid (HL) leakage during transport is a crucial risk management tool in the planning of transport of dangerous goods. It also enables the creation of comprehensive information systems that monitor the transport unit in real-time, indicate the presence of significant habitats along the transport route, and draw attention to possible threats, in particular to the health and lives of people and the environment. The main result of the presented research is the application of a computational model for determining the parameters of the dangerous zone in case of HL leakage and its graphical plotting along the transport route, estimating the probability of impacting the selected place by leaking HL. The model application results are presented in the form of calculated frequency of impacting the set of points in the vicinity of the HL transport route. Defined standardized frequencies of HL infiltration above a specified limit in liters per square meter in the event of leakage of the entire volume of HL from a road tanker (leaked volume of 30 m$^3$) form the basic set of information for creating relevant risk maps near busy traffic routes and subsequent selection of ecologically and spatially optimal routes.

**Keywords:** environmental impact model; transportation; hazardous liquid; dangerous zone; standardized frequencies of infiltration; sustainable development

## 1. Introduction

The global goal of modern and sustainable society reduction in the share of used and transported dangerous substances and goods has not been achieved to date. For this reason, the transport of dangerous goods is a permanent source of threats, not only to human life and health but also to other biotic components of the environment [1,2]. Researchers' attention in the field of specifics and risks of transport of dangerous goods (DG) in terms of basic physical properties and state of the commodity is most frequently divided into three main areas-transport of hazardous liquids [3–5], transport of hazardous gases (HG) [6,7] and transport of radioactive material [8,9]. Research in each of these areas is highly specific. The principles and methods of environmental impact assessment are often diametrically opposed, considering the specific physicochemical properties, the hazard of the substance to biotic components, methods for estimating exposure, and the extent of their negative effects. Researchers systematically investigate and seek solutions to reduce the risk and minimize the impact of accidents and other incidents during HL transportation.

The authors' research focused on the transport of hazardous liquids and modeling the consequences of accidents involving ADR vehicles with the subsequent leakage of HL into the environment. The aim of the article is to theoretically clarify the prerequisites, principles, and proposed procedure for computer modeling and quantification of the environmental impact of HL leakage on the environment in the vicinity of the transport route. The model is based on the HL spread in the terrain at the point of leakage on the transport route, considering the spatial parameters of the terrain, the influence of surface vegetation, and the properties of the subsoil at the point of leakage.

## 2. Theoretical Background

The basis of every research activity is the collection, sorting, and verification of input data and information. With regard to the research focus, the available international and national statistics were primarily used. The research applied mainly electronic data from EUROSTAT databases [10] and national statistical databases such as the Czech Statistical Office [11] and Statistical Office of the Slovak Republic [12]. In addition, other relevant sources, selected on the basis of an in-depth analysis of knowledge in the area, were included. Within the implementation of data mining, the primary goal was to apply up-to-date, valid, and available information.

Another significant part of the research activity was an in-depth analysis of available and relevant scientific work, focusing on environmental risk assessment during hazardous liquid transportation. The examined problem is multidisciplinary in nature. It was necessary to proceed from solving the physical principles of fluid spread, and the research into individual, social, and environmental vulnerabilities to exploring the possibility of increasing the resilience of transport systems and their surroundings. The authors continuously addressed these areas, for example, in a journal and book [13–19].

One of the most significant works in the addressed area is [15], which focuses on individual and social risk during the transport of hazardous substances. It presents research into transport risks of hazardous materials focused on mortality and personal injury. Attention is paid mainly to the theoretical framework of the negative impact of explosions, fires, gas dispersion, and bandwidth for liquids. The section on the bandwidth of liquids describes the process of possible leakage of liquids from the tank, its spread on the road, and seepage into the surrounding soil without considering its evaporation. The three physicochemical processes depend on a number of parameters of the leak site and its surroundings, for example, air temperature, soil water absorption, wind direction and speed, terrain slope, subsoil permeability, amount of leaked liquid, method of leakage, type of hazardous liquid, and others.

From the point of view of environmental sciences, it is necessary to address the impact of HL leakage and to propose measures to minimize the negative impact, especially biotic components of the environment. The research presented below focuses on the leakage of HL from the tank during its transport and its subsequent spreading (overland flow), infiltration, and evaporation into the environment near the place of leakage (Figure 1) [20].

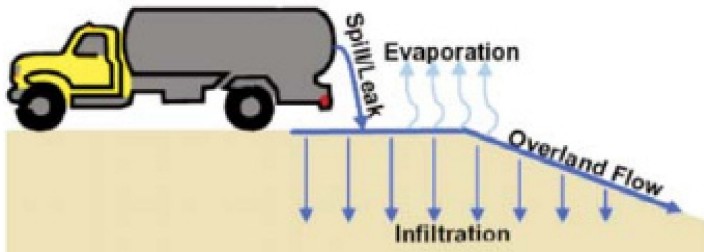

**Figure 1.** Scheme of processes by liquid outflow from car tank [15].

The impetus for the research presented below was the study focused on a risk analysis system for the transport of hazardous material [21]. The authors described in detail the risk

of multimodal hazmat transport, presenting relevant scenarios—fire pools for flammable liquids and evaporating pools for toxic liquids. For this, the appropriate impact distances for roads were selected: flammable liquid = 50 m, toxic liquid = 200 m, and the basic parameters for the calculations were defined. The estimation of the local probability of occurrence C can then be determined by the relation

$$C = \frac{Y_{loc}}{Y_{tot} * \frac{L}{L_{tot}}} \tag{1}$$

where $C$ is the local probability of occurrence (–), $Y_{loc}$ is the number of accidents on the segment of length (number), $Y_{total}$ is total number of accidents in the transport network (number), $L$ is the length of the segment for which the locality parameter is calculated (m), and $L_{total}$ is the length of examined transport network (m).

The work [22] that studied decision support systems for assessing the risk involved in transporting hazardous material was also crucial to the research. The article contains a detailed analysis of existing decision support systems-HAMER, HAZMAT, TrHAM, TrHazGis, TRAT-GIS 4.1, and GIIS (Global Integrated Information System). Links to international regulatory standards were cited as a key factor for risk management, e.g., European Agreement concerning the international carriage of Dangerous goods by Road (ADR), including Annex A and Annex B [23]. Links to parallel developed information systems, Graphical User Interface, Geographical Information System (GIS), Data Base Management System, and Multi-Criteria Decision Aid were also presented as significant. Decision support systems, most frequently developed as tools to support risk assessment and aimed to assess the level of risk, are described in more detail, for example, in [24–26].

Another inspiring output is the work of [27], focused on a model for analyzing the risks involved in the transportation of hazardous goods for the geographical information system. The article contains a description of suitable risk calculation methods, specifying the magnitude of the probability and possible consequence in a particular place of transport.

In the study focused on modeling the environmental risk of transporting hazardous materials by road, see Figure 2 [28], the authors explored the division of hazardous substances into groups-flammability, toxicity, reactivity, and oxidation. The criteria for targeting the research were environment, population, and road. They recommend evaluating the frequency of occurrence into five classes of very low (0–0.20), low (0.21–0.40), medium (0.41–060), high (0.61–0.80), and very high (0.81–1.00).

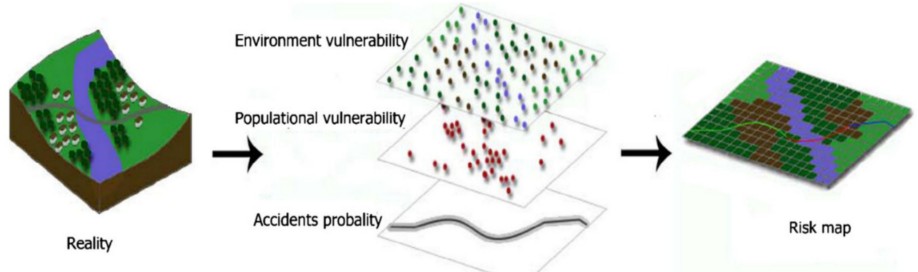

**Figure 2.** Scheme of the creating environmental risk map [28].

The main result of this research is the recommendation to define as many probable scenarios of expected leakage of hazardous liquids as possible. It is necessary to include individual scenarios in the map and gradually add other layers to them, such as population, vegetation, fauna, quality of soil, etc. In conclusion, the authors correctly state that the detailed multi-criterion analysis performed in this way is a solid basis for a software tool that is reasonably accurate in terms of quantity and quality of information sources and able to simulate real-life conditions and situations with the required accuracy.

The third area with the availability of relevant information sources is the outputs from solved scientific projects. A significant source is the CORDIS database, where the results

of scientific projects solved with the support of the European Union are presented [29]. From the point of view of the authors' research focus, the inspiration was the project TRANSrisk—Transitions pathways and risk analysis for climate change mitigation and adaptation strategies (2015–2018), Grant agreement ID: 642260 [30]. The main aims and objectives of the TRANSrisk project are to create a novel assessment framework for analyzing costs and benefits of transition pathways that will integrate well-established approaches to modeling the costs of resilient, low-carbon pathways with a wider interdisciplinary approach including risk assessments.

The project GRACE—Integrated oil spill response actions and environmental effects (2016–2019), Grant agreement ID: 679266 [28] was also stimulating. One of the project goals was to assess the impacts on the biota of naturally and chemically dispersed oil, in situ burning residues, and non-collected oil using biomarker methods, and to develop specific methods for the rapid detection of the effects of oil pollution. The sNEBA tool will be developed to include and overarch the biological and technical knowledge obtained in the project, as well as integrate with operational assessments based on expertise on coastal protection and shoreline response [31].

At the national level, several projects published on the website of the information system of research, development, and innovation of the Czech Republic [32] were proven beneficial. They contained records of 52 currently running programs to support science and research, information on 16 ongoing public tenders, 6311 projects in progress, and a total of 1.8 million records of the results and outputs of completed and progressing projects.

In terms of the article focus, the BIOTRA project, solved in 2008–2011 [33], was a valuable inspiration. Its main goal was to propose a procedure for assessing the impacts of the transport of dangerous goods on natural ecosystems and to create a support tool for selecting a transport alternative with a minimum environmental impact and predicting environmental damage to biotic components near the transport route.

## 3. Literary and Information Sources

Quantification of threats and risks generally involves the probability of an adverse event and its consequences. The recommended procedures for risk assessment in the prevention of major accidents [34] and in the national conditions of solvers, in particular the legal environment [35], are also based on this approach. The risks of stationary sources of hazardous substances and the resulting threats to the population are methodically elaborated in considerable depth [36,37]. Determining the level of risk for mobile resources is considerably more complicated. The risk assessment of mobile sources of hazardous substances with regard to the threat to the environment and its selected biotic components has, to date, relied only on qualitative or quantitative assessment of the impacts of the hazardous substance leakage into the environment. However, the evaluation carried out in this way is not sufficiently objective, and its results are difficult to compare for the individual biotic components. This aspect significantly complicates the subsequent decision-making processes.

The authors intend to present the proposed method of computer modeling and quantification of the negative impact of accidents associated with the release of hazardous substances of a liquid nature on the environment in the vicinity of the transport route. One of the research results is a model of the spread of the liquid escaping in an accident over the terrain near the place of leakage. Petroleum products and industrial acids were selected as representatives of hazardous liquids with regard to the quantities transported. In the available literature and projects, there are solely rough propagation models for large oil spills into water sources [38] or models of their impact on marine ecosystems; considerably less attention is paid to inland fauna and flora [39].

The presented approach and the proposed theoretical model allow us to calculate the parameters of flow, infiltration, and evaporation of the liquid and determine the degree of impact on the surface biotic components of the environment. The proposed theoretical model is supplemented by a module for serial calculation of liquid leaks at various points in

the route and a module for statistical processing of results based on probabilistic principles. The output of the model is the frequencies of impacting partial areas along the route by the hazardous substance in the liquid state.

The aim of the article is mainly to clarify the proposed method of assessing the effects of HL leakage, and to present the developed software tool for the assessment of environmental risks and their impacts on the transport of dangerous goods on land transport infrastructure. Attention is focused mainly on selected biotic components of the environment around the transport route (soil, vegetation, groundwater, and other objects). The acquired risk levels, the expected dangerous zones, and the frequency of hitting points around the route are the data needed to create a further developed software tool for deciding on the selection of environmentally friendly transport routes and bypasses of significant habitats in the defined area [40].

## 4. Materials and Methods

The materials and methods section presents the materials and methods used in the research, focusing on specific information sources and methods.

### 4.1. Materials

The groups of transported liquid hazardous substances were determined with regard to their expected negative impact on the environment and the volumes of individual types of transported HL and traffic accidents associated with their leakage. Based on the analysis of statistical data and discussions with experts, two groups of substances and their representatives listed in Table 1 were selected for the planned experiments and leak modeling.

**Table 1.** Groups of hazardous liquids and their representatives.

| Group of Substances | Typological Substances | Representative |
|---|---|---|
| Flammable substances | Petroleum products | Diesel, Petrol |
| Corrosive substances | Inorganic acids | Hydrochloric acid, Sulfuric acid |

The above selection of typical representatives of transported HL covers a significant part of the spectrum of traffic accidents with leakage of hazardous substances that occurred in the Czech Republic and the Slovak Republic. During leakage accidents, the effects of these substances on biotic components were demonstrated, and were largely irreversible.

The impact of each of these groups of substances or their representatives is invariably determined by many circumstances and conditions that can have a positive or destructive effect on the environment. In addition to the technical properties and failure conditions of the tank, the rate of spread of the contaminant is determined, for example, by current weather, terrain surface material and properties, vegetation cover density, geological subsoil permeability, location of vulnerable elements (adjacent dwellings, streams and groundwater, and infrastructure), but also the speed, technology, and effectiveness of the intervention immediately after the accident.

During the HL leakage into the soil, the most significant parameters are:

1. Slope of the terrain and the possibility of accelerated drainage;
2. Porosity, type, and composition of affected soil and its ability to absorb spills; type of soil microflora;
3. Aeration and soil moisture (depending on previous precipitation and current weather);
4. Current weather (particularly the temperature, wind direction and speed, humidity, precipitation intensity);
5. HL exposure time to soil (time interval between the occurrence of the accident and the arrival of the soil remediation group, and the method and type of remediation intervention).

The selected types of HL for the presented research are able to significantly influence the biogeochemical cycles in nature (hydrological cycle, hydrogen cycle, nitrogen cycle, oxygen cycle, and phosphorus cycle). The selected HL cause changes in the microbial flora, which plays a decisive role in the ongoing biogeochemical cycles in the soil. In such processes, the acidity of the medium (pH) and its redox potential (pE medium) change and, in addition, particular chemical reactions of escaped HLs with the environment occur at a specific site [40].

### 4.2. Hazard Zone and Risk during HL Transporting

The standard procedure for determining the hazard zone in the event of an HL leak involves an approximation of the shape of a pool of spilled liquid by an elliptical surface. However, such a procedure is sufficient to determine the bandwidth of the hazard only in slightly rugged terrain. The advantage of the approach is low computational complexity, whereas the disadvantage is the restriction only to the surface of the affected area, without the possibility of considering the infiltration process and its parameters. This fact significantly complicates the calculation and reduces the accuracy and objectivity of the estimation of the HL propagation parameters after its leakage.

The approach based on defining the terrain model near the leak site and dynamic consideration of the fluid progression within the theoretical model is more objective. The model formulated in this way monitors the balance of fluid in the field and below the surface and allows the prediction of data on the overflow of HL into surface waters. Within the research, models of three primary elements were topologically characterized:

1. Environment at the site of the event-necessary to describe the spread of fluid: The description of the medium contains an elevation and topographic model, which was improved several times. It was mainly an automated treatment of data discrepancies adopted from various map materials (e.g., the corner of the building located in the river). At the level of the required accuracy of calculations, this proved to be a significant problem;

2. Current physical parameters necessary for research into the dependence of fluid propagation: A total of 10 field experiments were performed. Based on these, it was possible to decide on computational algorithms, but also on the settings and format of input parameters to the model. The sensitivity of the model to the required fineness of spatial and temporal description was tested;

3. Uncertainties associated with the solution at a specific location: The objectification of uncertainties was based on stochastic modeling, especially with respect to their variability over time (e.g., soil saturation with water) and in space (e.g., details in uneven terrain and liquid capture on vegetation). The problem was solved by entering the interval values of individual quantities for probabilistic calculations.

### 4.3. Theoretical Principles of Used Models

The process of fluid propagation is discretized in the model of the selected space using a square network. Time discretization was performed by calculation in predefined time steps. The area of interest within the particular area is divided by the model through a square network. The surface of each element is considered horizontal and homogeneous in terms of properties. It represents a place where ongoing events are simulated in time steps, and a calculated balance of the leaked liquid is performed.

The altitude of the element is primarily determined from the elevation data. The basic database of geographical data of the Czech Republic (ZABAGED) is a digital geographical model of the territory of the Czech Republic [41,42]. The ZABAGED database contains two types of data that describe the morphology of the terrain-contour lines and dimensioned points.

The terrain model describes a part of the earth's surface in the vicinity of the considered HL leakage site. It is always represented by two files, one of them expresses elevation, and the other topographic data. Basic elevation data-contours and dimensioned points

were adopted from the ZABAGED database. First, values were assigned to objects with elevation data from GIS. The altitudes for the other objects were then recalculated using interpolation methods.

The identical procedure was followed in the topography. The types of surface objects to be displayed were specified (roads, watercourses, buildings, type of land cover-meadow, forest, arable land, and other areas). For each type, a specific group of physical properties that are needed to calculate the spread of fluid across the surface is defined. An example of building a terrain model is demonstrated in Figures 3–5. Figure 3 depicts a section of a map of the selected area of interest in the ZABAGED environment.

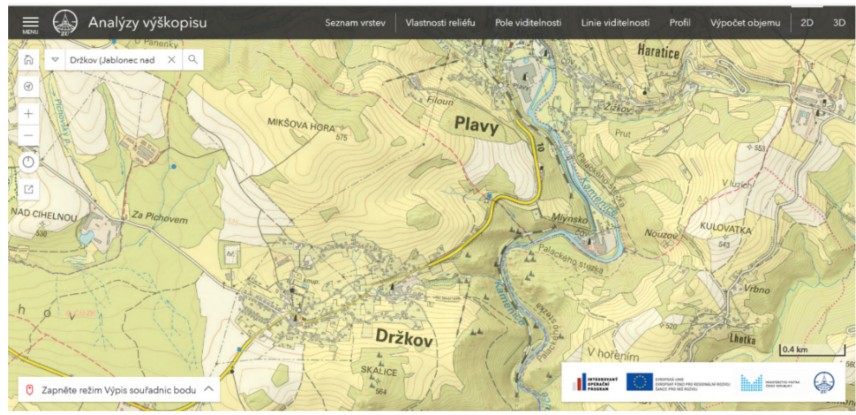

**Figure 3.** Map section of the selected area of interest (ZABAGED).

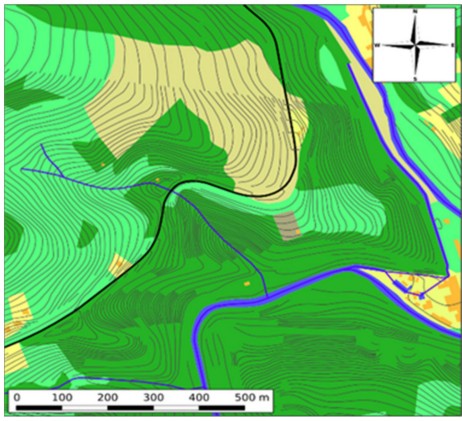

**Figure 4.** Contour map of the area of interest (ZABAGED).

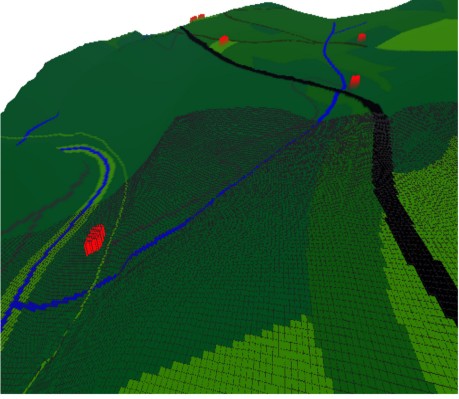

**Figure 5.** Terrain structure as a network of objects-view from the west (ZABAGED).

Figure 4 contains a section of the map with contour lines. Figure 5 demonstrates the structure of the terrain as a network of objects. For the sake of clarification, roads are marked in black, and watercourses in blue. In the views of the terrain, the buildings are marked in red. The contour map is simplified, and the buildings are not shown in this case. The results of the application of the model and the performed computational analyses are further demonstrated in the article on the given reference example of the selected area.

The fluid propagation model was developed using an algorithm for calculating fluid motion according to [43]. In each element of the selected square network, the change in the volume of the liquid is calculated in a given time step and a new level is determined. Changes in the volume of liquid in the element depend mainly on the following events:

1. Liquid overflow from one element to another;
2. Trapping of liquid on the earth's surface;
3. Infiltration below the surface;
4. Evaporation of the liquid.

Source of the force for fluid overflow between objects is the difference in level. The level is a function of time for each object. If no liquid has flowed into the building, then the level is equal to the height of the terrain. The flow rate of a liquid between elements is determined by:

$$v = k.\sqrt{\frac{H_i - H_0}{a}} \tag{2}$$

where $v$ is the liquid outflow rate (m/s), $k$ is the liquid movement coefficient (m/s), $H_i$ is the altitude of the liquid level in the object from which the liquid has drained (m), $H_0$ is the altitude of the liquid level of the object into which the liquid has flowed in (m), and a is the length of the side of the object (m).

The flow of the effluent is determined by the relation

$$Q = v.a.h = v \cdot a \cdot V_A \tag{3}$$

where $Q$ is the flow rate of the flowing liquid (m$^3$/s), $h$ is the height of the liquid pool layer (m), and $V_A$ is the volume of liquid on the surface, based on the unit area of the object (m$^3$/m$^2$).

For the total liquid balance on the element at time t from the beginning of the leakage

$$V(t) = [V_A(t) + V_B(t) + V_V(t)] \cdot a^2 \tag{4}$$

where $V$ is the volume of liquid bound to the element (m$^3$/m$^2$), $V_B$ is the volume of liquid infiltrated below the unit area of the element (m$^3$/m$^2$), $V_V$ is the volume of liquid evaporated on the unit area of the element (m$^3$/m$^2$).

The change in the volume of liquid on the surface of the element over time $\Delta$t is then determined by the expression

$$\Delta V_A = \Delta V_i - \Delta V_0 - \Delta V_B - \Delta V_V \tag{5}$$

where $\Delta V_A$ is the increase in the volume of liquid on the surface (m$^3$/m$^2$), $\Delta V_i$ is the increase in the volume of liquid that flowed from adjacent elements (m$^3$/m$^2$), $\Delta V_0$ is the decrease in the volume of liquid drained into adjacent elements (m$^3$/m$^2$), $\Delta V_B$ is the decrease in the volume of liquid infiltrated below the surface (m$^3$/m$^2$), and $\Delta V_V$ is the decrease in the volume of evaporated liquid (m$^3$/m$^2$).

Figure 6 schematically illustrates the procedure for applying a fluid propagation model to a single planar element.

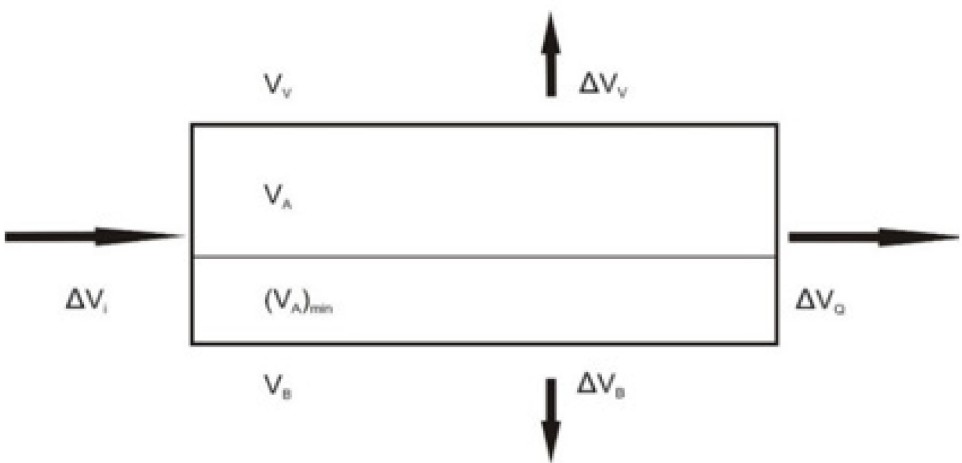

**Figure 6.** Scheme of fluid motion balance on a planar element during one time step.

The volume that flows out of the element during the time $\Delta t$ is

$$\Delta V_0 = Q.\Delta t \tag{6}$$

where $Q$ is the flow of the outflowing liquid (m$^3$/s) and $\Delta t$ is the time during which the change in the volume of the liquid that flows out of the element is monitored (s).

If the volume of liquid on the surface of the element $V_A$ is smaller than the volume of liquid retained by the surface $V_{Amin}$, the losses $\Delta V_0$ and $\Delta V_0$ always equal zero.

Liquid retention on the earth's surface occurs when the liquid can no longer move. Characteristically, liquid can always flow only to adjacent elements on a lower level. If a lower level occurs in two or three elements, the current will be split. The volume $\Delta V_0$ is then given by the sum of the volumes of all adjacent elements. If none of the adjacent elements has a lower level, the liquid will not drain from the element in a given time step. It is an element or group of elements corresponding to the local depression of the terrain, which is gradually filled with liquid; only then can an overflow occur. A special case is the elements on the bank of the river or water reservoir. No further overflow is considered for these elements. Here, there is only a summarized transboundary amount that enters surface waters.

Liquid infiltration is a practical consequence of the fact that during the spreading of the liquid on the surface, it also penetrates below the surface of the terrain. The rate of infiltration depends on the permeability of the soil and the viscosity of the liquid. The level of water saturation of the subsoil can also play a significant role in the infiltration process.

Infiltration in the model is described using a simplified Green-Ampt infiltration model [20]. According to this approach, the volume of liquid infiltrated per unit time on a unit area is determined by the relationship

$$f = K\left(1 + \frac{P\psi}{F}\right) \tag{7}$$

where $f$ represents the volume of liquid infiltrated per unit time on the unit area (m/s), $K$ is the hydraulic conductivity of the medium for the given liquid (m/s), $P$ represents the ratio of the volume of unsaturated pores (–), $\psi$ is the suction pressure height of the medium below the surface (m), and $F$ is the volume so far infiltrated per unit area (m$^2$). A more detailed description of the Green-Ampt model can be found in, e.g., [43].

Amount of evaporated substance per time step $\Delta t$ depends mainly on the specific type of substance. Evaporation reduction takes precedence over other lowering effects as it occurs at any temperature, regardless of the level, also in the liquid retained on the terrain

and its cover. The relationship for evaporation used in the model is based on the Yellow Book methodology [41] and has the form

$$\Delta V_V = 9 \cdot 10^{-4} \cdot C \cdot \frac{MW \cdot P}{T + 273} \cdot \frac{1}{\rho} \Delta t \tag{8}$$

where $\Delta V_V$ is the amount of vaporized substance (m3), per time step $\Delta t$ (s), $MW$ is the molecular weight (kg/mol), $P$ is the saturated vapor pressure (Pa), $T$ is the characteristic liquid temperature (°C), $\rho$ is liquid density (kg/m³), and $C$ is the pool area factor (–), the value of which decreases with increasing pool and increases with wind speed. When calculating the balance on an individual element, neither the size nor the shape of the pool is known. We do not consider the wind speed here. Therefore, $C = 1$ applies in our model, as due to the relatively short evaporation time in the movement of the liquid, we assume that the error resulting from this simplification will be insignificant.

Based on the evaluation of the above characteristics, it is possible to evaluate and quantify the following state characteristics for each area element in the selected time intervals:

1.  Amount of free liquid (capable of outflow);
2.  Level height;
3.  Saturation of capture capacity (or unused capacity);
4.  Degree of subsoil saturation.

### 4.4. Probabilistic Approach for Fluid Motion Modeling

Numerical values obtained by interpolation from elevation data or assigned values according to topographic data enter into deterministically formulated algorithms of fluid spreading. The result is exact numbers, indicating the values of the state parameters for the individual area elements at a specified time.

The altitudes of most planar elements are calculated using interpolation that is random. This finding means that if we repeat the calculation of altitudes, we constantly obtain different sets of altitudes. As the density of the height data increases, the differences between the heights determined in this way decrease. In this way, the fact that data on minor surface irregularities is then not available, and can only be estimated, is modeled. At the same time, however, such inequalities can fundamentally change the direction of propagation of the escaping liquid. The variability in the morphology of the terrain model is thus directly related to the variability in the calculations of the spread of the leaked liquid.

The information above indicates that with the number and quality of substrates describing the morphology and characteristics of the surface over which the liquid spills, a deterministic approach based on a single variant of the terrain model has almost no objective informative value. Therefore, a stochastic, probabilistic approach was selected [44].

The chosen approach is based on the extension of the deterministic approach by the repeated implementation of the calculation of terrain model parameters and the corresponding calculations of fluid propagation with randomly selected parameters using the well-known Monte Carlo method. For each variable, specific rules for generating their values are specified, usually the lower and upper bound and the type of distribution. In our reference case, a uniform and triangular probability distribution was used [45].

The calculation procedure consists of two main steps. In the first step, a series of terrain models is generated. In the second step, for each terrain model, a series of simulation calculations are performed to determine the fluid propagation parameters for the retention and infiltration capacity parameters according to the terrain models in step 1. Both steps are performed for the specified number of variants.

Using the Monte Carlo method, it is possible to obtain value data sets of monitored quantities for each surface element at any time, including level height, amount of free liquid, filling of capture capacity, amount of infiltrated liquid, or amount of liquid flowing into specific surface elements. Statistical analyses are subsequently performed on the

acquired data sets, focusing on selected factors related to the computational estimation of the risk level, such as:

1.  The probability that the liquid will reach the relevant element within a defined time from the moment of leakage. As with one element, this value can be specified for an entire group of elements. Such a group may represent, for example, a line representing the bank of an endangered watercourse, or an area of rare habitat, etc.;

2.  The probability of exceeding the monitored limit values, for example, a limited infiltrated amount posing a serious threat to the soil or groundwater. By analogy, even in this case, the determined probability can relate not only to one, but to the whole group of elements.

The main result of the implementation of this procedure is an estimate of the frequency distribution of the surface, the infiltrated amount of HL, the amount of liquid that has touched the watercourse or water reservoir, etc. The results obtained represent the basic input data for quantifying environmental risk.

## 5. Results

The initial tests were performed on a deterministic model. All input model parameters were entered directly and fixedly. The calculations were focused on the fluid spreading in the field under different ambient conditions. The area of the individual pools and their shape were assessed depending on the choice of infiltration characteristics and the rate of spreading, including the time evolution and possible leakage into surface waters. The above procedure was also applied in the implementation of the first series of field experiments. They aimed to identify such parameters in which the results of computer simulation of individual spilled pools were closest to the results of field observations and experiments, and it was possible to assess the sensitivity of the models to changes in individual input parameters.

### 5.1. Software Tool

According to the calculation based on probabilistically defined principles, it is possible to compile maps of the probability of hitting individual surface elements around the route and maps of the probability that a certain amount of liquid has been infiltrated in a particular element. The probability maps obtained in this way are a suitable basis for the calculation of ecological damage, and it is possible to confront them directly with maps of specific habitats in the vicinity of the transport route. Alternatively, it is also possible to estimate the degree of damage to the affected habitats according to the intensity of the negative impact, as well as to estimate the probability that such an event will occur.

An algorithm was designed for the implementation of practical calculations and, subsequently, a software tool was developed [46].

The model is implemented in a SW package consisting of several separate applications that are created in the Delphi development environment. Applications (preprocessors) represent user support of the model. The first application performs calculations of fluid propagation from a point source for randomly generated surface variants. The second application takes over the results of the previous application and calculates the required statistics from them (e.g., an estimate of the probability of the individual places being hit by the liquid). The third application estimates the statistics of hitting positions in the vicinity of the transport route. It takes over the results of the second application for individual point sources of leakage on the transport route.

Using the first application, input data is entered into all three of the above applications. The second application is used to display the results of the calculation in 3D. For this purpose, the OpenGL graphics library working with 3D graphics scenes, which is part of the Windows operating system, was used.

The input data required for the calculation are transmitted to these applications in a text table file in CSV format and SHAPEFILE format, describing the required geographic

data. The results of the calculations are transmitted in the form of a special format designed for this model.

The main program has two basic modules, computing and display. The calculation module ensures the creation of a terrain model and the calculation of the above-mentioned quantities characterizing the state of fluid distribution in individual elements. The display model allows viewing the results in 2D/3D mode, creating image documentation and exporting the results. An example of a software tool display window is demonstrated in Figure 7.

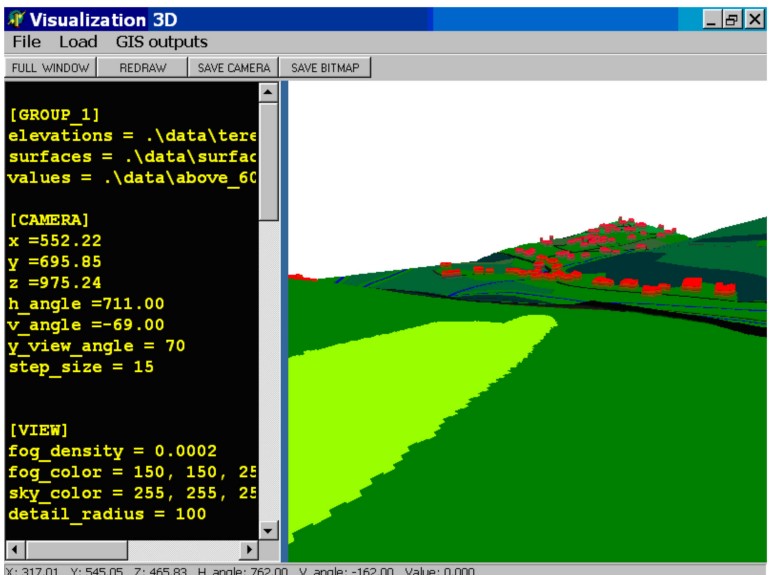

**Figure 7.** Software tool display window-terrain model.

From the user's point of view, it is significant to mention subprograms for compiling input files-describing the elevation and topography of a specific area element. These files contain information of various types:

1. Identification of source data files;
2. Placement and rendering of objects;
3. Requirements for modification of element heights;
4. Physical parameters of the element.

Depending on the nature of the problem and local conditions, it is possible to define up to several dozen different objects for one calculation. It only depends on the need to distinguish various topologically related objects, e.g., first, second, and third class roads, railway corridors, buildings, watercourses (streams), and water areas (rivers and reservoirs). According to the data, which are usually available in digital form, it is possible to distinguish, for example, different habitat types and other specific areas and objects [47].

*5.2. Dependence of the Pool Area and HL Penetration into Surface Waters on Infiltration Parameters*

Modeling HL propagation and its parameters was performed in the place of the gorge with a stream, located 0.5 km SW of the village Plavy. A total of three escape points were chosen at the edge of the road above the slope. Figures 8 and 9 demonstrate the modeling areas of the pools obtained and the amounts of HL that flowed up to the watercourse. Figure 8 shows the areas of the three pools considering three infiltration intensities. The amount of liquid retained in the terrain was in all cases at the level of 5 mm/1 m$^2$. To illustrate the possibilities of the software tool, Figure 9 shows the amount of liquid that would reach the stream from the individual pools to a minimum level of infiltration.

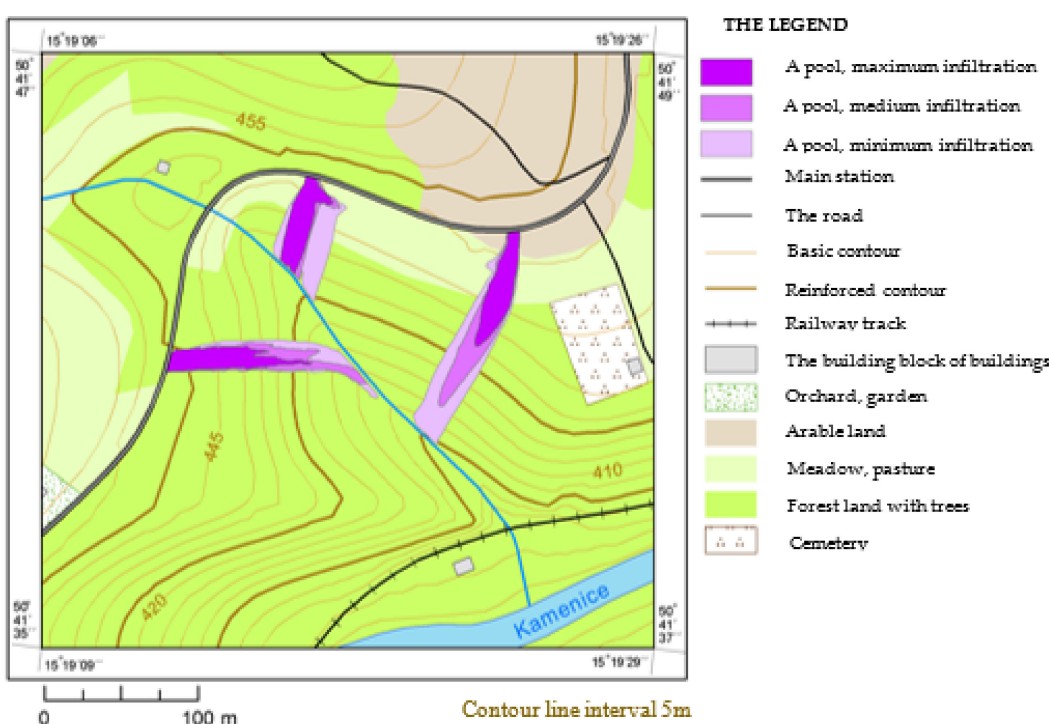

**Figure 8.** Dependence of the extent of the pool on the intensity that enters the stream.

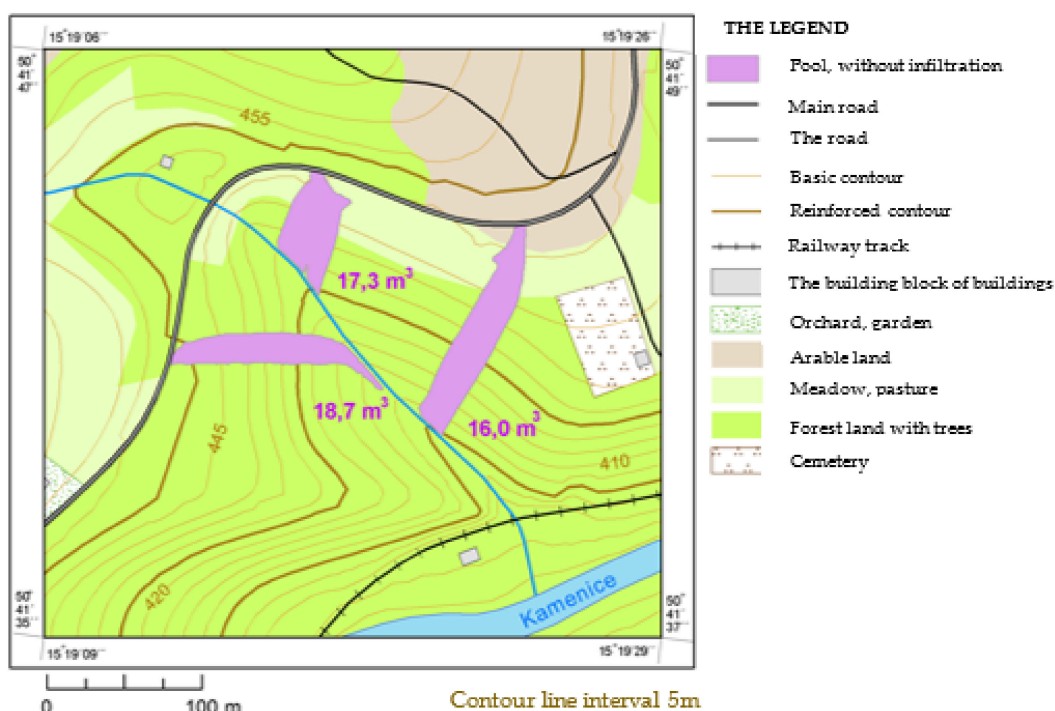

**Figure 9.** Pools without infiltration-volume of liquid, infiltration.

### 5.3. Liquid Spreading Time Course

The research also examined the development of a pool that is formed when 30 m$^3$ of liquid is poured from a tank for 5 min. Figures 10–13 specify the level heights above the ground at times of 100 to 400 s from the moment of HL leakage from the source. The level decreases with increasing time from the leakage. Completion of HL leakage from the tank is expected within 300 s. However, the pool is expanding further. The last presented

state at 400 s represents the final state-stopping the flow of the pool, as the entire volume of liquid is already retained on the terrain cover.

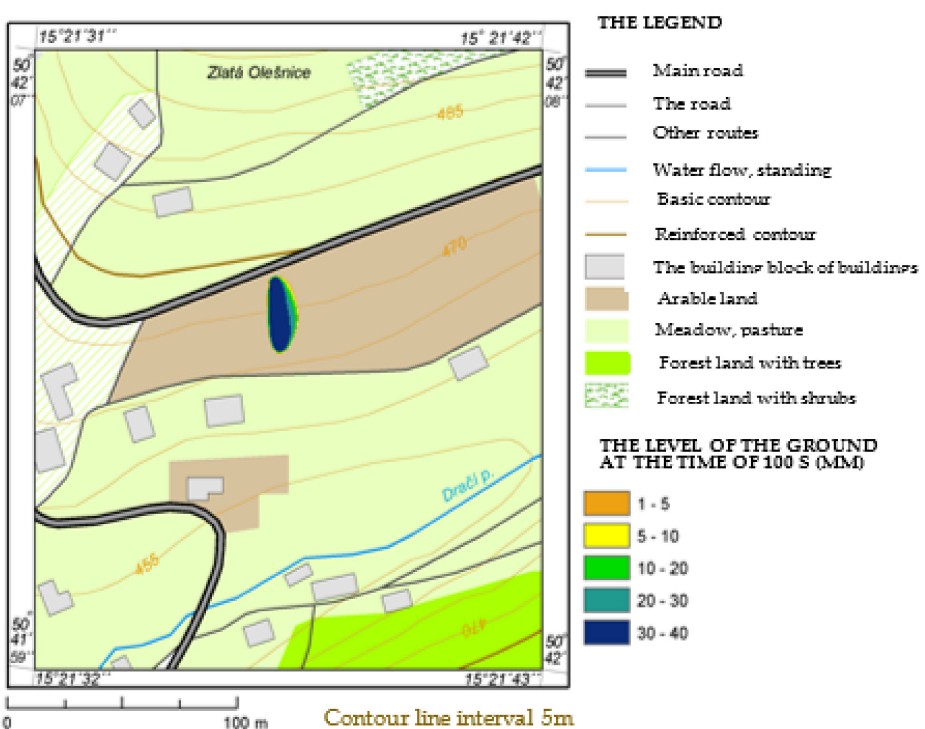

**Figure 10.** Height of the pool level above the terrain level in t = 100 s.

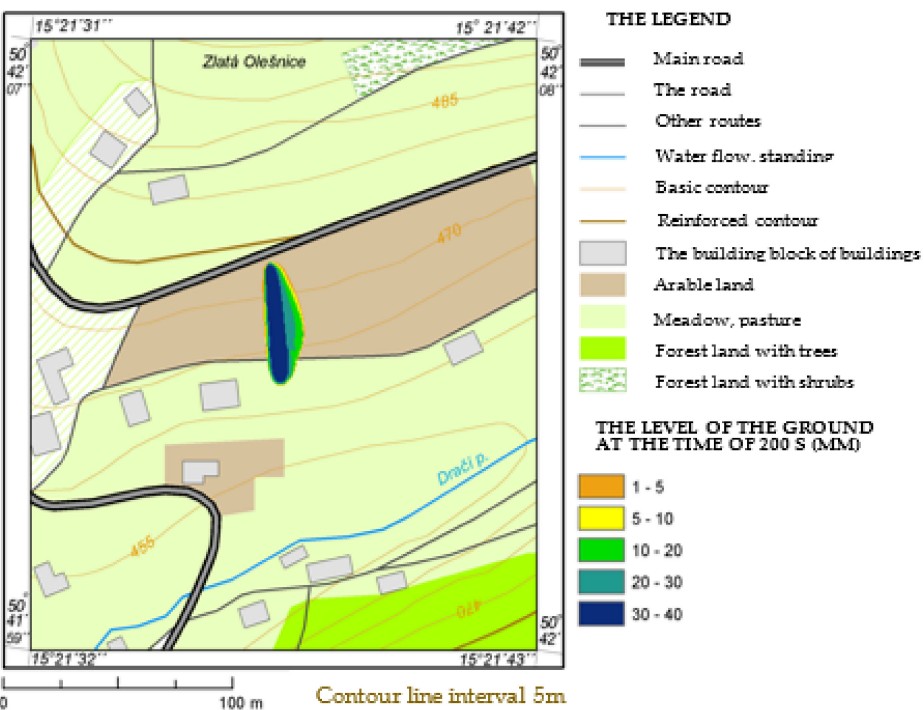

**Figure 11.** Height of the pool level above the terrain level in t = 200 s.

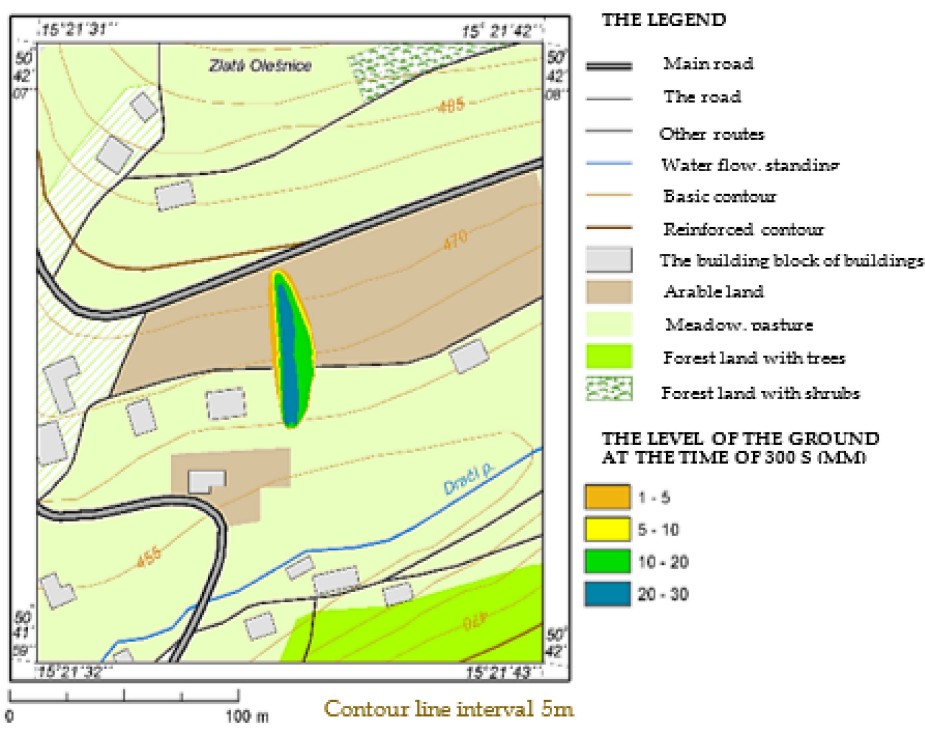

**Figure 12.** Height of the pool level above the terrain level in t = 300 s.

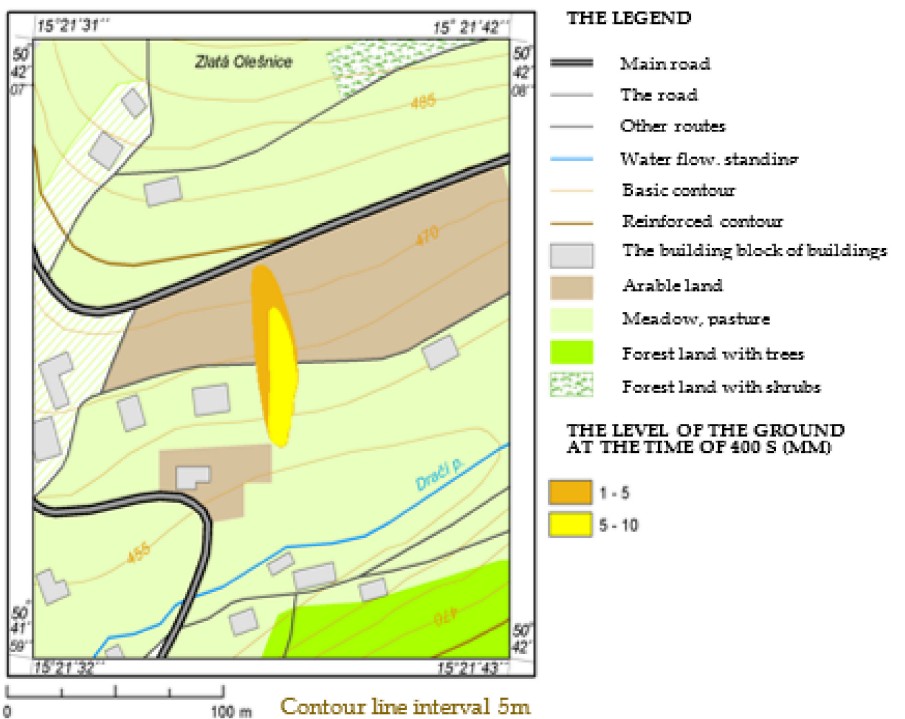

**Figure 13.** Height of the pool level above the terrain level in t = 400 s.

Figure 14 shows a map of the isolines of the distribution of the infiltrated liquid (diesel) after the end of the leakage. The amount of leaked liquid is specified in mm, which corresponds to values in $l/m^2$.

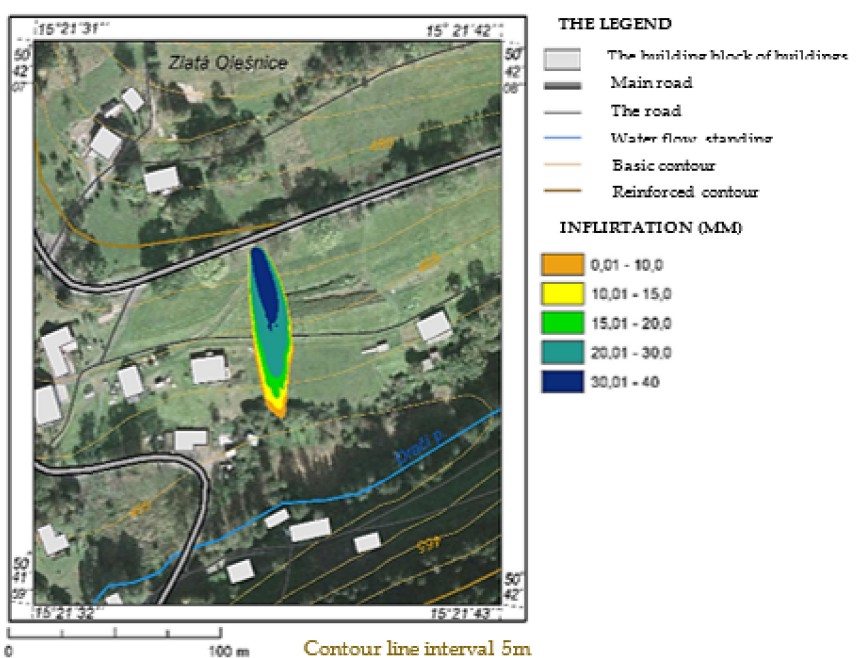

**Figure 14.** Final distribution of infiltration in the event of a leakage of 30 m$^3$ of diesel.

*5.4. Rendering of the Dangerous Zone*

The drawing of the dangerous zone is one of the most significant results of the proposed procedure for modeling the process of spreading the leaked liquid. The hazard zone was plotted as the total area of the pools in the individual surface elements. Figure 15 demonstrates an example of the dangerous zone during the transport of 30 m$^3$ of diesel in the section of the road E65 Loužnice–Držkov.

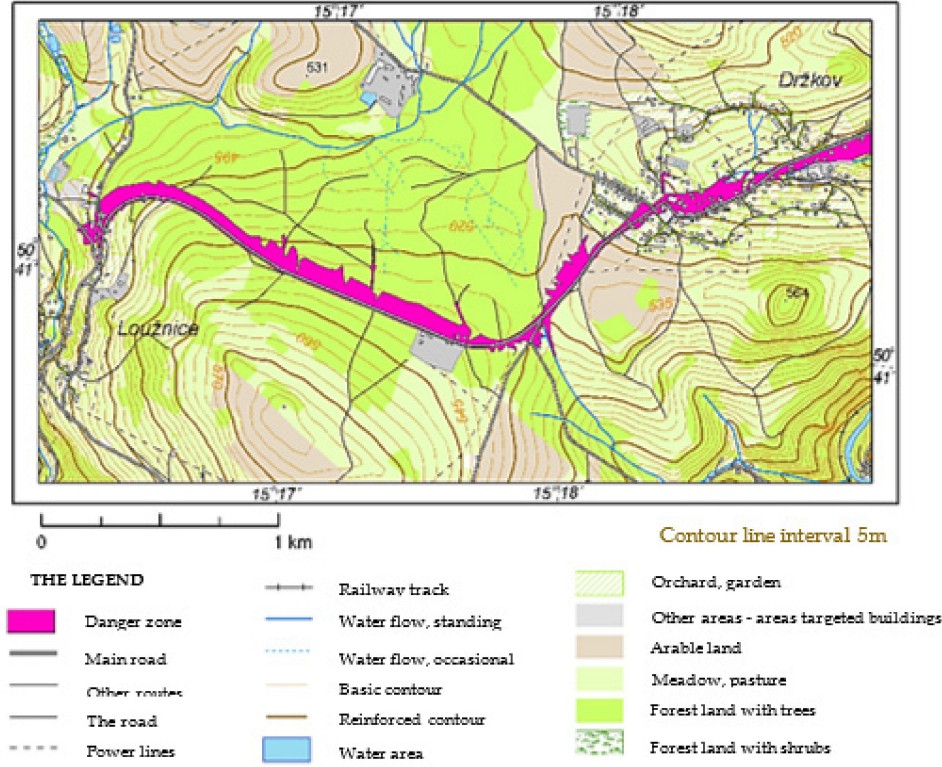

**Figure 15.** Dangerous zone in the road section E65 Loužnice–Držkov (transportation 30 m$^3$ of diesel).

*5.5. Probability of Impacting a Point (Area Element) with Liquid*

From the point of view of evaluating the negative effects of HL transport on the components of the environment, it is significant to be aware, for example, of the probability of impacting a specific point by escaping HL. To illustrate the graphical possibilities of displaying the results, a map of the probability distribution in the event of an HL leakage at the point where the road bypasses the local terrain wave can be observed in Figure 16. At such a point, the liquid can spread in different directions, depending on the terrain. As such local deviations are not evident from the available map data, the uncertainties in the model had to be solved by a stochastic approach.

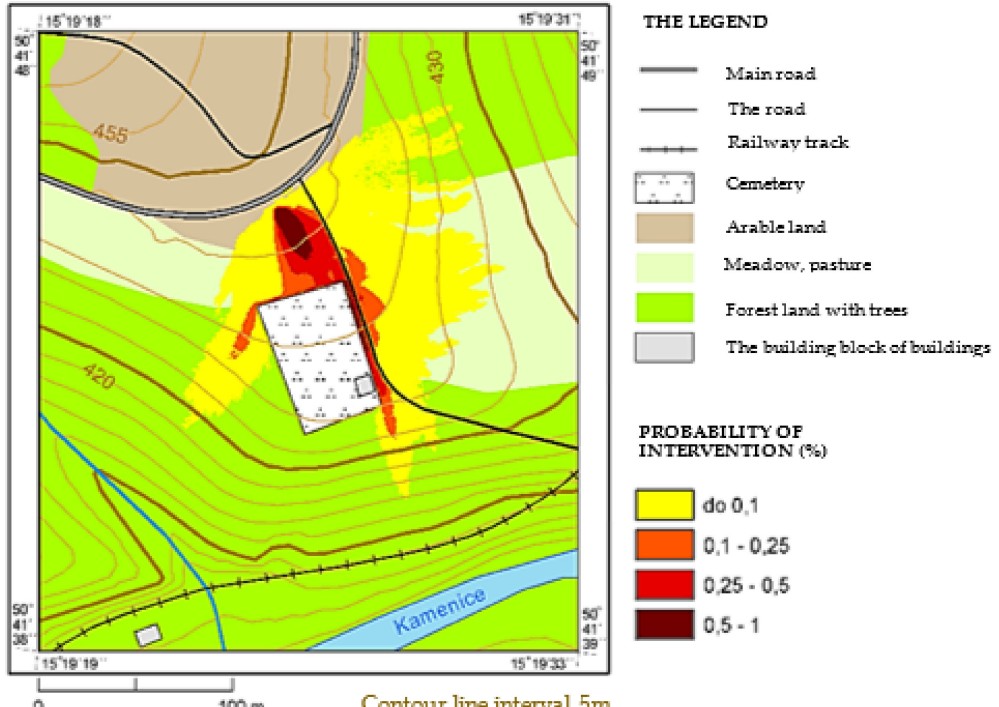

**Figure 16.** The probability of the points being impacted by the leaked liquid in the vicinity of the site of the considered HL leakage.

*5.6. Threat Zone and Its Characteristics*

A schematic representation of the proposed theoretical model for the computational determination of the hazard zone with its characteristics is demonstrated in Figure 17. The main characteristics of the hazard zone are considered to be the frequency of impacting discrete points near the route.

1. Infiltration of fluid into the soil;
2. Pool depth;
3. Seepage of liquid into surface waters.

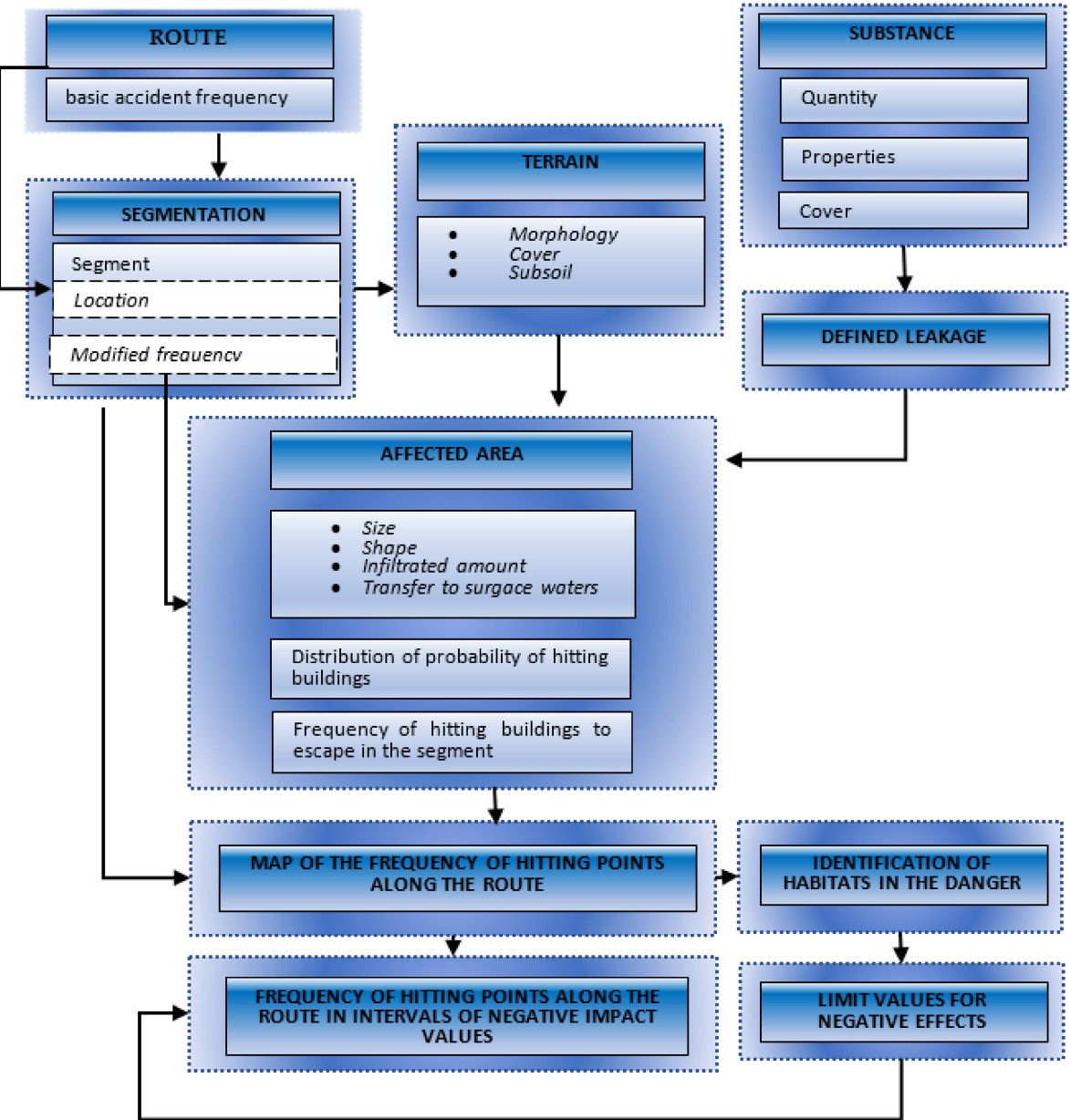

**Figure 17.** Scheme for determining the frequency of impacting the points near the route by negative influence.

*5.7. Procedure for Determining the Frequency of Impacting the Points near the Route*

The initial data is the quantity and type of transported HL. The most unfavorable variant of the accident assumes a one-time leakage of the entire amount of HL, which determines the input parameters of the considered leakage L. The next step of the procedure for determining the shape and extent of the dangerous zone is the division of the route or its selected section into segments and the determination of possible points, i.e., places of accidents with subsequent leakage of HL. For each point, the HL spread is calculated for the amount of leaked liquid and the type of liquid transported. A section of terrain in which the liquid can spread is selected for each segment. The following is a stochastic calculation of fluid spread with a random selection of elevation, topography parameters, and fluid motion characteristics within specified limits and consideration of a suitable statistical distribution. The result of the calculation is the field of probability of impacting individual points in the event of a leakage in the selected segment, as demonstrated in Figure 18.

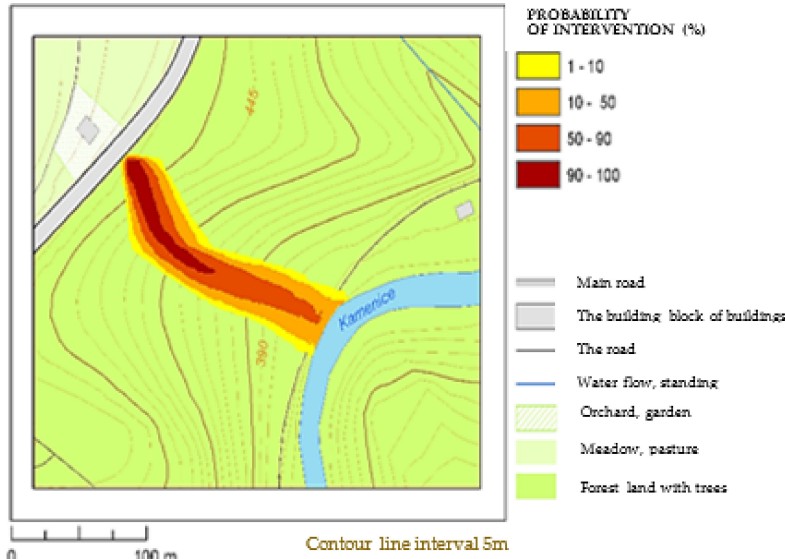

**Figure 18.** The probability of impacting the points near the route (leakage of 30 m³ HL at the side of the road).

When calculating the leakage from each potential source, the probabilities of partial areas being impacted are multiplied by the relative accident frequency for a given kilometer of the route. With a profile spacing of 10 m and one point on the profile, it is 100 points per 1 km of the route. If three points per 10 m are selected, there will be a total of 300 points per 1 km of the route. The accumulated values are referred to as standardized frequencies of impacting individual points, i.e., frequencies corresponding to one accident per 1 km of the route for a specified period of time (usually one year).

### 5.8. Frequencies of Impacting A Set of Points near the Route

Figure 19 depicts a detail of the dangerous zone in the vicinity of the reference section—the section of the Class I road between the municipalities of Držkov and Plavy with the stated standardized frequency of the affected areas in the immediate vicinity of the section. The frequencies were normalized with respect to one accident with HL leakage per 1 km of route.

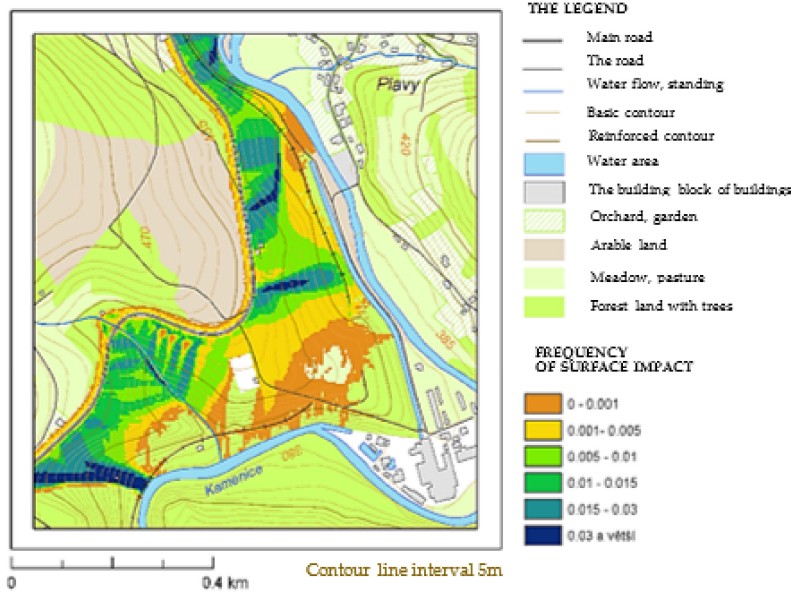

**Figure 19.** Standard frequency of impact on surfaces in the event of a HL leakage of 30 m³.

Figure 20 highlights the impact frequency isolines for three negative impact intensity categories. It was found that the largest share of the frequency of impacting areas near the route falls in the middle category of infiltration intensity (interval 2–10 L/m$^2$).

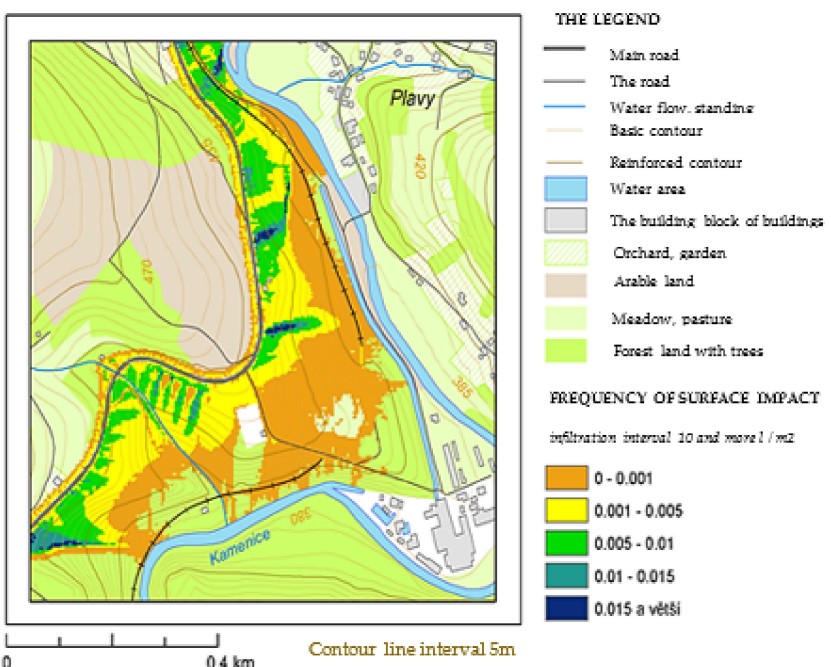

**Figure 20.** Standardized frequencies of infiltration of surfaces by infiltration above 10 l/m$^2$ in case of leakage of 30 m$^3$ HL.

As part of the research and the verification of the model functionality, other scenarios with different environmental conditions, HL, leakage extent, etc., were gradually implemented. The results of the research created a suitable theoretical basis and conditions for the future application of fluid spread models in complex information systems that can be instrumental in optimizing the selection of an environmentally safe route of NL transport. The aim should be to minimize the environmental impact of a possible HL leakage on the environmental components around the transport route.

## 6. Discussion

The framework for environmental impact modeling for the transport of hazardous liquids presented in the article is a multidisciplinary problem. The complexity of the solution should consider all interests at the global level (the crucial aspect), at the national level (reflecting the level of maturity and culture of the state), at the regional level (affecting every citizen, animal, and plant), at company level (taking into account the focus specifics), and at the individual level (everyday behavior) [48]. The presented framework for environmental impact modeling for transporting hazardous liquids is the basis for the discussion in scientific and professional circles, which should gradually lead to changes in the legal framework. Such a discussion requires a valid scientific basis, long-term experience and a more dimensional view of the solution of HL transport and its negative impacts.

Any transport of dangerous goods poses a significant risk to the environment. In every traffic accident, there is a leakage of HL. When transporting large volumes of HL in tanks, the risk to the animal and plant communities at the scene of the accident is unacceptable. Therefore, when planning the transport, it is necessary to consider the possible effects of HL leakage into the environment and the possible damage or destruction of its biotic components. The following uncertainties need to be discussed:

1. The amount of HL, usually limited by the maximum volume of the transport packaging (typically tanks);

2. Leakage time, limited by the time of complete emptying of the transport packaging; the worst-case scenario, the immediate leakage of the entire volume of HL, is usually assumed;

3. A place of leakage which is not known in advance and may, with varying probabilities, be located anywhere on the HL transport route;

4. After the liquid has been spread onto the earth's surface, three basic physical processes occur at different intensities: overland flow, infiltration, and evaporation.

These events must be considered and thus modeled simultaneously, as both infiltration and evaporation depend mainly on the area of the pool formed, and conversely, the flow is influenced mainly by the loss of liquid that was absorbed or evaporated.

As demonstrated above, the flow of leaked HL is generally dependent on the properties of the fluid, the slope of the terrain, and the type of surface. As the slope increases, the width and depth of the pool decrease by default. The depth of the pool is significantly dependent on the type of surface. For example, on a relatively smooth surface (asphalt or concrete), the typical depth is given in millimeters. In the places with vegetation cover, the depth of possible retention increases appreciably (in centimeters). The local inequalities, e.g., furrow in the field, drainage and catch channels on roads, etc., can be even more significant. The infiltration is conditioned by the permeability and porosity of the substrate. Evaporation is conditioned primarily by the properties of the liquid, the ambient temperature, but also the speed and direction of the wind.

Another important circumstance in the event of HL leakage in the field is the ability of the surrounding subsoil to absorb HL, to form closed pools, or to allow seepage into surface or groundwater. However, the issue is so complicated that it is beyond the scope of the article to discuss all possible alternatives to leakage, spread, and impacts of HL on biotic components in the vicinity of the transport route.

The main goal of the article is to present selected results of the authors' research in the field of evaluating the impacts of HL leakage on biotic components in the vicinity of the transport route was fulfilled with the following activities:

1. Clarification of the theoretical framework of the proposed method of modeling the spread of HL in the vicinity of the transport route;

2. Application of a theoretical apparatus for predicting the magnitude and intensity of action in the event of an HL leak during transport-dangerous zone;

3. Presentation of selected functionalities and possibilities of a software tool for stochastically based models of forecasting the spread of liquids in the terrain.

The solution makes it possible to evaluate the ecological risks associated with the transport of HL on the road infrastructure in a pre-selected area. The original benefit of the research is the possibility of assessing the impact of HL leakage on biotic components of the environment in the vicinity of the transport route, focusing on the dependence of the pool area and the penetration of HL into the subsoil. The significant variables include the infiltration parameters, the HL leakage rate, the size of the affected area, the time course of the HL spread, the probability of the point (an area element) being impacted by the liquid, the dangerous zone, and its characteristics.

The future direction of research is to automate the quantification of possible ecological impacts on biotic components of the environment using 5G network technology and the application of the Internet of Things. This solution will allow the selection of the optimal transport route for predefined areas with regard to possible environmental impacts. Optimization criteria will need to be set according to priorities, e.g., the protection of fauna, flora, or entire biotic communities in a given area [49,50].

The connection of several scientific disciplines with the application of modern information and communication technologies is a prerequisite for a future fully automated and functional information system aimed at minimizing environmental risks not only in the transport of HL, but also other dangerous substances and goods. The future development of the society following the sustainable development principles must become a primary goal. Environmental protection is closely linked to the protection of human health. In the

HL transport, it is necessary to gradually meet the goals of sustainable development [51], so that every human activity is carried out in accordance with the principles of environmental protection. Meeting the 17 goals and 197 targets defined by the United Nations is the only way to protect the environment across the planet.

The examined problem enables the risk quantification if the magnitude of the consequences of the leakage of HL on individual components of the environment is also quantified. The HL leakage tends to affect more biotic components of the environment at the same time. The recipients of the consequences, and thus the risk, are the individual plant and animal organisms and, subsequently, human society. Therefore, it is necessary to find agreement in the valuation of impacts on different recipients. These consequences need to be converted to a common base, quantified by a single quantity. The quantification is practically possible only through the monetary valuation of damages (monetary value of risk). However, risk quantification requires elementary agreement to assess the consequences. It is a crucial issue, as the consequence appreciation depends on the value that a particular community attaches to the risk recipients.

Another problem in risk assessment is determining the acceptable risk value. The applied community values can vary significantly, not only between different states but also within one country. It follows from the above that in the processes of environmental risk assessment it is necessary to involve not only experts in the field of natural sciences, but also socially and economically oriented disciplines.

## 7. Conclusions

The resulting solution is a procedure for determining the zone of danger to the environment with its characteristics in the form of the frequency of impact of discrete points/partial areas near the route by the negative influence of a certain intensity. By negative influence, we mainly mean the infiltration of liquid into the soil, the depth of the pool, including the liquid retained on the terrain surface, and the transfer to surface waters where the amount of liquid is not evaluated at individual points, but in the entire coastline.

To calibrate the selected model parameters, field tests were performed with water spilled over the terrain from the fire tank. The model uncertainties were solved by stochastic calculation with pre-selected intervals of individual quantities respecting their determined probability distribution. The model can calculate the standardized frequency of impact of partial areas near the transport route, where one point can be impacted from several potential points of leakage.

The problem solution specified the relations describing the flow, infiltration, and evaporation of liquid in actual terrain. Their algorithmization enabled the development of a software tool for the subsequent calculation of the spread of the leaked liquid. The software tool was gradually improved with the possibility of generating various morphological details, adding modules for compiling and importing input files, capturing the properties of defined types of objects in the modeled area, as well as appropriate user environment modifications.

One of the main outputs is the generation of maps of the frequencies of the affected areas near the road within the given limits of the intensity of the negative impact in a potential accident with HL leakage in different route segments.

It is the basis for further research into the overall ecological damage in terms of a comprehensive quantification of the consequences for human lives and health, as well as animal and plant habitats. The developed model of fluid spread in the terrain was an original contribution of the researchers.

Research into individual areas of security leads us to new challenges on how to increase the level of protection of society. Within our research teams, we have repeatedly encountered the limits of what is possible and feasible. New issues are emerging as new challenges to security and sustainable development are addressed. Research into the sustainable development of society requires teamwork with the representation of researchers from various disciplines. If society is to continue to develop sustainably, it is

necessary to link the results of all relevant research areas and look for synergistic solutions that will bring a new quality of life.

**Author Contributions:** Conceptualization, Z.D., B.L. and P.F.; methodology, P.F.; validation, M.B. and P.F.; formal analysis, Z.D. and B.L; investigation, P.F. and B.L.; data curation, P.F.; writing—original draft preparation, Z.D., B.L. and L.M.; writing—review and editing, B.L., L.M. and P.F.; visualization, M.B.; and supervision, Z.D.; All authors have read and agreed to the published version of the manuscript.

**Funding:** This research was funded by Internal Grant of UNIZA: Evaluation of fire-technical characteristics of natural and synthetic (including recycled) organic materials used in transport, ID 12716 and VEGA Assessment of the level of resilience of key elements of land transport infrastructure, ID 1/0159/19.

**Institutional Review Board Statement:** Not applicable.

**Informed Consent Statement:** Not applicable.

**Data Availability Statement:** All data used were correctly cited in the article with references to their sources. As part of the case study, some information was obtained through consultations in practice and was transformed into a written form. Some of the information used for security research of specific objects is sensitive. Information that is confidential or sensitive is not included in the article.

**Acknowledgments:** Internal Grant of UNIZA: Evaluation of fire-technical characteristics of natural and synthetic (including recycled) organic materials used in transport, ID 12716 and Ministry of Education, Youth and Sport Czech Republic: Methodology for assessing the impact of transport routes on biodiversity and environmental components, ID 2B08011.

**Conflicts of Interest:** The authors declare no conflict of interest.

## Abbreviation

| | |
|---|---|
| ADR | Agreement concerning the international carriage of Dangerous goods by Road |
| ZABAGED | Basic database of geographical data of the Czech Republic |
| DG | Dangerous Goods |
| GIS | Geographical Information System |
| HG | Hazardous Gases |
| HL | Hazardous Liquid |

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
