# Peer review of "Environmental Impact Modeling for Transportation of Hazardous Liquids"

_sustainability, doi:10.3390/su132011367_

Round 1
Reviewer 1 Report
• On the first page (line 24) there is a fullstop before begginig of the sentence. It should be removed.
• The key word 'danger zone in case of leakage of hzardous liquids' is too long. It should be shortened or removed.
• Line 47. 'The article presents the authors' research focused ....' It should be formulated. Suggestion is to remove the part 'The article presents the'.
• The subchapters 1.1. Current knowledge and solutions in the field of research into the effects of HL leakage and 1.2. Theoretical framework and expected benefits of the solution should be removed from Introduction. This subchapters should form new chapter that should follow chapter Introduction. This chapter should be titled Theoretical background while second subchapter should be titled Literature review instead of Theoretical framework and expected benefits of the solution. Expected benefits of research should be listed in introduction.
• Line 252 The research indicated that the approach based on defining the terrain model near the leak site and dynamic consideration of the fluid progression within the theoretical model is more objective. It is not clear which research? Some previous or this one?
• In line 294 it should be backspace between words identical and procedure.
Author Response
The first reviewer
Thank you for your opinion and qualified comments and recommendations. We have incorporated all of them in full.
On the first page (line 24) there is a full stop before beginning of the sentence. It should be removed.
the error has been fixed
The key word 'danger zone in case of leakage of hazardous liquids' is too long. It should be shortened or removed.
the keyword has been abbreviated to danger zone
Line 47. 'The article presents the authors' research focused ....' It should be formulated. Suggestion is to remove the part 'The article presents the'.
“article presents" the has been deleted
The subchapters 1.1. Current knowledge and solutions in the field of research into the effects of HL leakage and 1.2. Theoretical framework and expected benefits of the solution should be removed from Introduction. These subchapters should form new chapter that should follow chapter Introduction. This chapter should be titled Theoretical background while second subchapter should be titled Literature review instead of Theoretical framework and expected benefits of the solution. Expected benefits of research should be listed in introduction.
subchapters 1.1 and 1.2 have been changed to chapters 2 and 3 with new titles, at the same time all the following headings and subheadings were renumbered
Line 252 The research indicated that the approach based on defining the terrain model near the leak site and dynamic consideration of the fluid progression within the theoretical model is more objective. It is not clear which research? Some previous or this one?
“The research indicated that” was deleted
In line 294 it should be backspace between words identical and procedure.
The backspace has been filled in line 294

Reviewer 2 Report
The present paper is very well written and deserves publication as it covers an issue that is of interest in the research community. It should be improved on the following to cover better the potential reader:
1) In the introduction part more references on relevant technical software that deal with environmental modelling should be provided
2) In section 2 the references for the infiltration model and the evaporation model used should be provided
3) Please check in both sections 2 and 3 the equations used (there are some type errors) and the captions as they have been moved in parts of the text
4) Some more references on the chosen probabilistic approach in section 2.4 are needed
5) Please include an abbreviation list of the abbreviations and the symbols used across the whole paper
Author Response
Thank you for your review and qualified comments and recommendations. We have incorporated all of them in full.
The present paper is very well written and deserves publication as it covers an issue that is of interest in the research community. It should be improved on the following to cover better the potential reader:
- In the introduction part more references on relevant technical software that deal with environmental modelling should be provided
References 18-20 were added to meet the reviewer's request. Environmental Modeling & Software, Environmental Modeling, Software and Decision Support, Environmental Protection Agency.
- In section 2 the references for the infiltration model and the evaporation model used should be provided
References 20, 44 and 45 have been added.
- Please check in both sections 2 and 3 the equations used (there are some type errors) and the captions as they have been moved in parts of the text
Thanks for notifying us, formal deficiencies in the equations have been fixed
- Some more references on the chosen probabilistic approach in section 2.4 are needed
References 44 and 45 have been added as key sources.
- Please include an abbreviation list of the abbreviations and the symbols used across the whole paper
The list of abbreviations used was included in the article for conclusion.
